# Fair Development Transition of Lignite Areas: Key Challenges and Sustainability Prospects

Christina G. Siontorou

Laboratory of Simulation of Industrial Processes, Department of Industrial Management and Technology, School of Maritime and Industry, University of Piraeus, 185 34 Piraeus, Greece; csiontor@unipi.gr

**Abstract:** As Europe transitions away from fossil fuel energy, the challenges associated with closing mines, along with restructuring mine-dependent communities, continue to reduce the rate at which this transition occurs. A large volume of research has been published during the last decade on the fair development context of mine reclamation. Using lignite mines as a case study and a properly modified analytical-hierarchy-process-based roadmapping methodological framework, the current study attempted to elucidate the key challenges and the bottlenecks that researchers view as critical to a just transition. The results indicated four critical areas that researchers expected to drive policymaking: social implications, stakeholder divergence, land-use challenges and sustainability prospects. Further, the coupling of mining operations to green strategies, such as circular economy and recycling, seems to be put forward by the academics as a viable solution to enhance the sustainability of the sector.

**Keywords:** lignite mines; mine closure; environmental and social impacts assessment; land reclamation; mine restoration

## 1. Introduction

Decarbonization has been strongly linked to environmental protection, sustainability, and climate change mitigation, as well as with social inequalities and fair transition policies. Greece has already set a goal of withdrawing lignite plants by 2028, aiming for 80% by 2024 [1,2]. The competent authorities have formulated a suitable plan for the energy transition, putting due emphasis on just transition development [3].

In the policymaking field, transition strategies with a focus on socioeconomics have been mostly developed during the last decade in an effort to address energy safety issues and high unemployment rates [4]. Notwithstanding, a just transition requires a strongly sustainable approach for successfully handling economic vulnerabilities, political ramifications and social disruption. Thereby, a sustainable transition policy framework should consider both the process and the results in society [5–7]. While technology covers a big part of the *process* theme, the *results in society* theme encompasses a quite diverse, multi-thematic, somewhat obscure and currently unforeseeable area, worthwhile of focused multidisciplinary research [8,9].

Built upon the explicitly stated views and opinions of the scientific community regarding the shaping of the *just transition* research domain, this work studies the key challenges and the bottlenecks that the relevant publications have clearly indicated as critical to fair policy development. Thus, this work goes beyond the presentation of an informed opinion about the fair development transition framework of lignite areas, by revealing how the science-base of the domain perceives, assesses, and weighs the non-technology obstacles to mine reclamation. Further, the coupling of mining operations to green strategies, such as circular economy and recycling, that seems to be put forward by academics—as a viable solution to enhance the sustainability of the sector—is a new concept with many rivalries. Although the results of this study need to be further verified by non-academic field players, they could facilitate the just transition to a more sustainable post-mine future.

## 2. Methodological Framework

The methodology used in this study has been previously developed by the author and extensively implemented in technology roadmapping via an analytical-hierarchy-process-based research evaluation framework. The analytic hierarchy process is a comprehensive analysis framework for multi-objective, multifaceted and multi-criteria-based knowledge sectors; it utilizes a simple hierarchy of concepts: objectives, criteria and alternatives. For technology roadmapping purposes, the lower level was replaced by internal barriers, i.e., technology drawbacks, limitations or disadvantages, highlighting the dominant technology trajectories and the viable emergent technologies within a research domain [10–12]. In this study, this methodology has been properly adopted to handle the evaluation of the strategy/management/policy sub-domain of lignite mine reclamation through a four-stage process, as described in Figure 1. To the best knowledge of the author, this is the first time that such a methodology has been employed to analyze the research trends of a policy-making process.

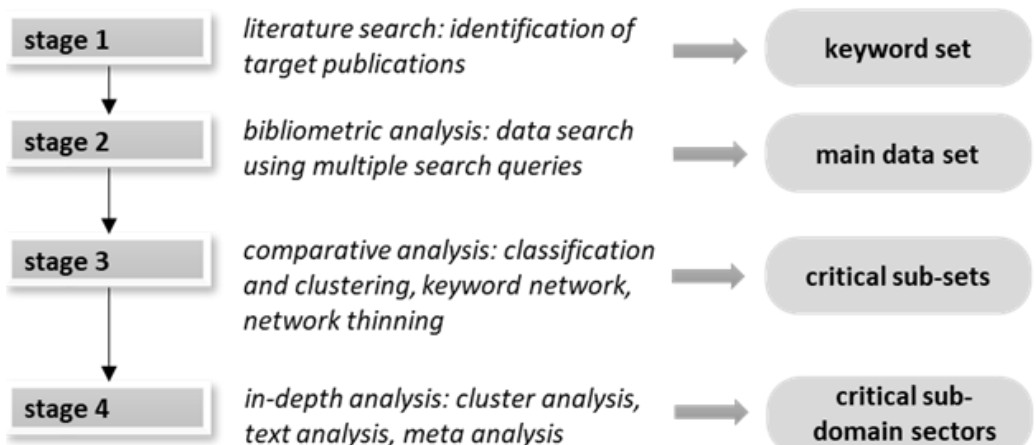

**Figure 1.** Overview of the methodological framework implemented in this study for elucidating the researchers' perspectives over the key challenges in fair development transition of lignite mines.

The first two stages deal with data acquisition and bibliometric analysis; once the target literature group was identified, the resultant set of keywords or search terms yielded the main data set that was subsequently split into subsets by means of comparative analysis. A cluster analysis was then used to structure the network of terms, in which each item was placed according to its significance. Finally, the resultant hierarchy of terms and concepts was manually validated and corrected, if needed. For more details, please refer to [10–12]. All software used herein was homemade.

### 2.1. Data Acquisition and Bibliometric Analysis

ScienceDirect was chosen as the data source for the literature collection. All the information was collected during April 2023. The survey on mine reclamation, technology exempted, covered the period 1993–2022, putting emphasis on the socio-economic frame after 2010 in order to highlight the domain dynamics that are expected to shape the forthcoming policy development. The data search strategy was adopted from the Mogoutov and Kahane [13] algorithm and merged with the semantics developed by Batzias and Siontorou [12]. Terminology limitations were avoided using multiple search queries: lignite mine areas, mining; mine closure, reclamation, post-mining, rehabilitation, restoration; land-use change, agricultural; shifting, transition, phase-out; circular economy, circularity; social, society, attitude, response, social movement; policy, socioeconomic; fair, just, etc. Although every effort has been made to produce a reliable and comprehensive data set, including manual checks and controls, the whole corpus of published works has not been identified herein. Energy shifting and just transition have received a lot of attention

lately, particularly after the hardships arising from the war in Ukraine, urging a variety of technology-oriented journals to blend policy views with technical discussions; thus, information mining and meta-analysis have been neither straightforward nor easy.

The number of papers retrieved by the ScienceDirect collection was 4898, 18% of which referred to the reclamation of lignite mines, mostly (ca. 52%) published within the last decade. Excluding reviews, editorials, books, chapters in books and conference proceedings to avoid double entries, the resultant data set included 422 papers. Double entries had to be evaluated and deleted manually.

### 2.2. Comparative Analysis and Clustering

The main data set was divided into subsets based on the comparative analysis methodology proposed by Lee and Su [14] for producing knowledge maps that highlight dominant and emerging knowledge structures. Briefly, the scientific input (cited articles) and the scientific output (article citations) formulated a cognitive network of keywords, terms or concepts, the weighted centrality of which—i.e., the number of associations with other nodes in any given knowledge level over the total sum of associations within this level—reflected the importance of the node in the network. It is generally accepted that if the centrality of a node is greater than 0.1, it means that the node is important in the whole network [15]; yet, since a lot of relevant data collected in this study, especially at lower knowledge levels, were included in technology-oriented papers and thus scored very low in comparison, network thinning was preferred. Based on the information–theoretical algorithm of Cheng et al. [16] and a cut-off value of 0.06, the analysis yielded the critical nodes (subsets) that form clusters representing the currently influential paths as well as the future trends.

The produced subsets indicated the critical sectors and the concepts within each sector that shape the specific sub-domain, i.e., the fair development transition of lignite areas. An in-depth analysis of each critical sector yielded a comprehensive review of the problems that the mining industry must successfully manage in order to enhance sustainability and social consensus. It is worthwhile to note that the scope of this paper was the presentation of the most justifiable generalizations and not the identification of the contingencies that make global generalizations impossible; thus, this stage of the current study followed the basic meta-analysis principles by coding the characteristics of each paper that might relate to the size of effects obtained in different papers as well as to the growth of the dominant network. Extensive manual checks and re-checks could not be avoided at this stage.

## 3. Results and Discussion
### 3.1. Trends in Publications, Clusters and Comparative Analysis

The most significant concepts regarding the centrality values are shown in Table 1. Social aspects, sustainability (as a general term) and stakeholders scored high, followed by land use and transition (as a general term).

**Table 1.** Top ten concepts/terms with respect to centrality values.

| Concept/Term | Centrality Value |
| --- | --- |
| Social aspects | 0.77 |
| Sustainability | 0.65 |
| Stakeholders | 0.63 |
| Land-use change | 0.43 |
| Transition | 0.32 |
| Landscape form | 0.23 |
| Management | 0.14 |
| Geoenvironmental analysis | 0.09 |
| Resource management | 0.07 |

It is worthwhile to note that the search context plays a critical role in the assessment of the significance of a concept within any given knowledge level. For example, *restoration* had a value just above the cut-off threshold (0.09), whereas *restoration technology*, found in a lower knowledge level within the cluster of land change use had a value of 0.14. *Geoenvironmental analysis* scored high within a technology-context search but quite low (0.09) within the policy-context search. Likewise, *resource management* was found just above the threshold value, although *resource depletion*, a strongly related concept term as per both economics and technology, scored higher at a lower knowledge level within the sustainability cluster. In the transition cluster, a sustainability subcluster was formed that was more significant than other concepts linked to the transition concept, such as climate change, activism, green economy, justice, benefit analysis, etc., that actually scored below the cut-off threshold.

The network produced is shown in Figure 2 as a cognitive map, clearly indicating four critical areas that researchers expect to drive policymaking: social implications, stakeholder divergence, land-use challenges and sustainability prospects.

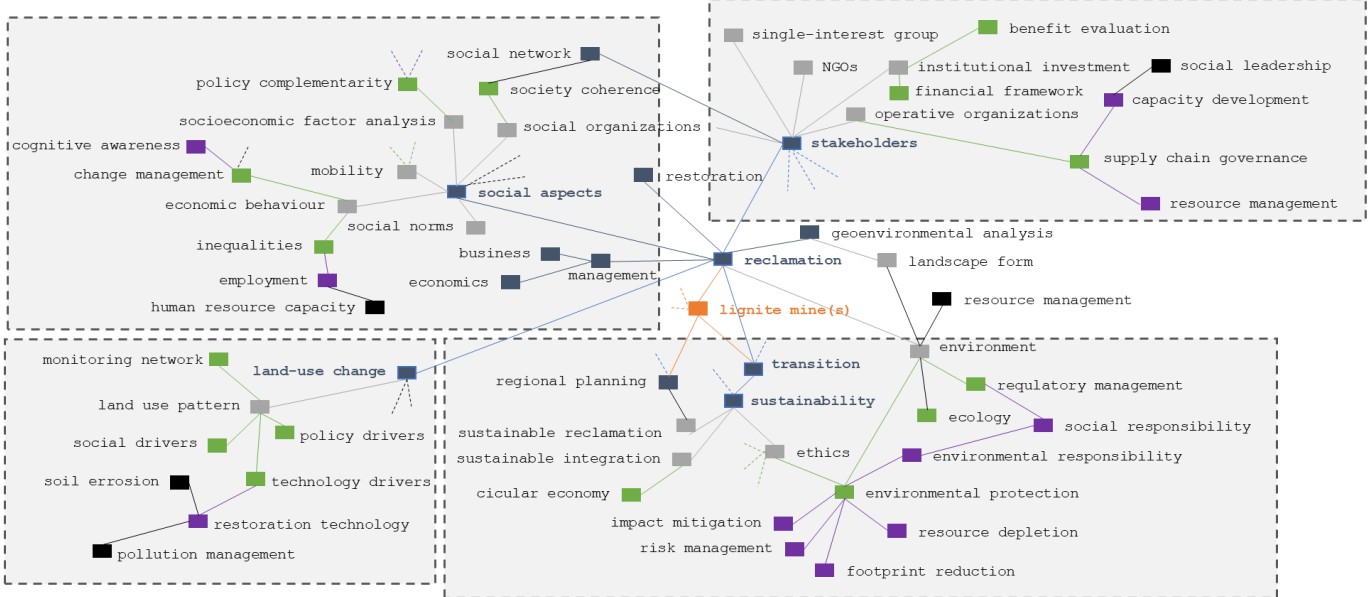

**Figure 2.** Extract from the upper levels of the cognitive map of the scientific sub-domains affecting bi-directionally lignite mine reclamation (technology exempted): social aspects (upper-left grid), stakeholders (upper-right grid), land-use change (lower-left grid) and sustainability (lower-right grid). The different layers are indicated by color, i.e., orange for level 1, blue for level 2, gray for level 3, green for level 4, purple for level 5 and black for level 6. The clusters have been formed around the nodes in bold. Dashed lines represent connections below the threshold point.

### *3.2. Policymaking Drivers*

3.2.1. Social Implications

The social network node formed a dense but rather restricted cluster covering the general socio-economic consequences of mine closure (Figure 2). Within the cluster, society (organizations, coherence and network) scored moderately (centralities of 0.22–0.25 within the cluster), yet higher than mobility (0.14), especially in papers published after 2018. It seems that workers' relocations were avoided in many cases (e.g., see [4,17]. Behavioral economics became more systematic after 2012, whereas two distinctive pathways appeared after 2016: (i) cognitive awareness and change management, mostly in engineering and/or engineering-linked management journals (e.g., see [17,18]), and (ii) inequalities and employment, mostly in social and environmental science journals (e.g., see [19–22]). Socioeconomic indices, such as economic performance, community hazards, environmental impact, etc.,

have been extensively proposed (e.g., see [19,22–26]) as a suitable guiding system to mine repurposing, attaching, however, great importance to policy complementarity (centrality of 0.13) with respect to investment and growth. Social norms presented a moderate to low significance (0.11), mostly linked to gender imbalances and economic inequalities.

A closer and more in-depth look at the evolution of the cluster in recent years showed a remarkable diversity over two critical aspects: (i) employment, displacement and relocation in post-mine communities (e.g., see [24,27]) and (ii) community attributes, acquired due to the operation of the mine, that hinder mine reclamation (e.g., see [20,22,28,29]). From a technological viewpoint, mine closure is part of the life cycle of the mining operation. Most importantly, mines are usually in operation for very long periods (50 years or more), giving ample time for both society—to structure a solid economy and culture around the mine—and mining companies, to degrade the environment under the tolerance of the society [27]. Several researchers believe that, despite the regulatory measures in force and effect and the socio-economic responsibility of the mining companies, the duration of the operation is such that the socio-economic indices shift dramatically towards a largely irreversible dependency of the social structure to mining: locality is defined by the mobility of the workers, the infrastructure and economy are developed to support the mining operation and its employees, career advancement is restricted to mine job opportunities, the health system is specialized to mine-related health risks, etc. (e.g., see [30–33]). This relationship is surprisingly bilateral: there exists a growing ethical consciousness in the mining industry, actually going beyond corporate social responsibility, that unused mines should be rehabilitated and returned to society as productive units.

Seen within the above context and through social studies, expert opinions, lessons learned and theoretical analyses, and the most prominent causal drivers to community-derived mine reclamation obstacles are summarized in Table 2. It is worthwhile to note that, as the published studies highlight, mine closure brings about several positive effects that could prove quite advantageous to policymaking and to the development of useful tools for just transition. The opportunities arising for other business activities—such as agriculture, tourism, education and crafts [20], the eminent environmental improvements that will be put into place during mine rehabilitation [34], the regained availability of resources (water, food, etc.) [18], the increase in property value [28], etc.—represent beneficial outcomes that, when integrated into a carefully designed and early implemented transition plan, could become the basis of fair post-mine development.

### 3.2.2. Stakeholders

The stakeholders' node (Figure 2) formed a significant cluster around governance (centrality value of 0.32 within the cluster), linked to resource management, capacity development and social leadership. The financial and investment frameworks, with centralities values of 0.23 and 0.22, respectively, recorded the second-best values, putting emphasis on the attributes of the post-mine projects that attract the attention of public and private organizations. Benefit evaluation has also scored relatively high (0.21), especially as regards the scope and effect of the repurposing plans. Non-governmental organizations (NGOs) and single-interest groups yielded moderate-to-low scores (0.12 and 0.09, respectively); their association with stakeholders was due to their negative impact on stakeholders' cohesiveness.

**Table 2.** The most frequent causal drivers to community-derived mine reclamation obstacles, as stated in academic publications in the period 2010–2022.

| Causal Driver | Frequency of Occurrence | Description [1] |
|---|---|---|
| Social risk mismanagement | 23.54% | Lack of timely and adequate community preparation for mine closure, resulting in economic and health stress. |
| Inappropriate training for career changes | 21.22% | Lack of appropriate training programs to fit the existing or developed employment opportunities. |
| Illegal occupation of abandoned mine structures | 21.08% | Devaluation of infrastructure that increases rehabilitation costs. |
| Major environmental degradation | 21.03% | Severe consequences of natural capital (especially water and soil) that increase rehabilitation costs. |
| Insufficient financial support and compensation | 17.11% | Lack of viable options, such as early retirement, debt counseling or long-term aid, resulting in rapid poverty increase. |
| Long-time gap between mine closure and reclamation | 13.44% | Community disintegration due to unemployment and/or mobility of the workers; thus, any attempts made to build resilience into the community are doomed to fail. |
| Ineffective measures and strategies to mine closure | 13.00% | Unrealistic timeline of the decommissioning plan that leads to unfair post-mine development, abrupt employment change and social disruption. |
| Lack of regulation of post-mine landscapes | 8.08% | Insufficient legislative framework for the disposition of the rehabilitated mine areas that leads to increased opposition between the society and the government. |
| Contested meanings of what is included in mine reclamation | 6.97% | Intervention plans that are not holistic lead to incomplete rehabilitation that lacks the required degree of specificity. |
| Administrative and budget constraints complicate planning and timelines | 5.16% | Limited capacity to implement the planned changes, thus considerably increasing delivery times and deliverables. |

[1] In the sense of connotations, knowledge associations and causal effects.

Most of the bodies of publications examined herein placed great importance on the influence of stakeholders on the fair development transition of lignite mines. As many researchers pointed out, mine stakeholders have divergent interests, objectives and socio-cultural backgrounds (e.g., see [20,22,35]); post-mine interested parties are even more diverse and could include all people or groups of people impacted by the mine decommissioning and reclamation (e.g., see [17,27,36]). Figure 3 presents the stakeholder subsets identified herein, actually forming two main groups: the affected/impacted group and the project contributors' group. Depending on the phase of the post-mine project, in the mine-dependent community structure, and the composition of the project enablers team, an individual can belong to more than one group(s) or/and move freely between groups throughout the post-mine projects.

Nonetheless, stakeholder identification should occur early in mine repurposing in order to ensure the suitability of the project outcome and the minimization of unnecessary changes (e.g., see [37,38]). Yet, the classification of stakeholders as internal (primary) or external (secondary) is often vague, especially in mine-centered communities. Internal stakeholders are usually the contributors to the rehabilitation project, and as such, they should be involved early; however, fitness-of-purpose is better judged by the external stakeholders (i.e., the affected/impacted group in Figure 3) that express the project requirements [39,40]. In the latter group, private owners are community members who are expected to be the users of post-mine infrastructure; their requirements are extremely important for the investment strategy, and their satisfaction is one of the determining factors for the success of the project (e.g., see [28,41–44]). Thus, external stakeholders should at least be consulted early in the project.

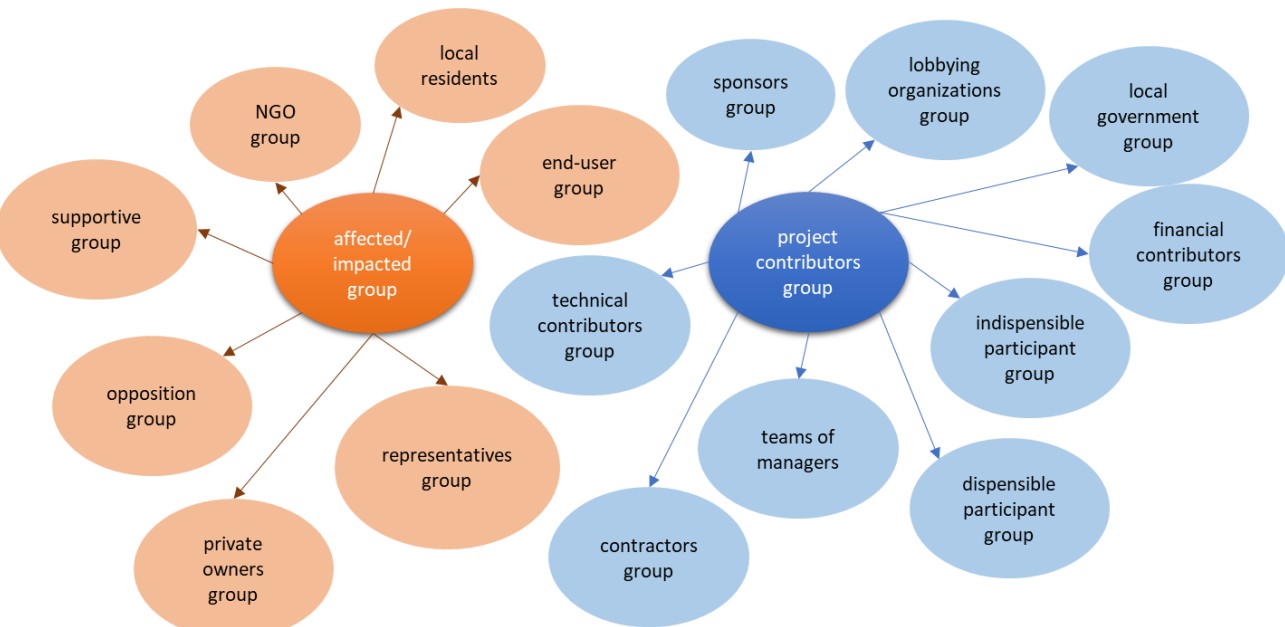

**Figure 3.** Interested parties in post-mine reclamation projects: the project contributors' group (blue) and the affected/impacted group (orange); classification is not exclusive, i.e., an individual can belong to more than one group(s) or/and move freely between groups throughout the post-mine projects.

In such a diverse group of interested parties, rivalry among stakeholders and competing stakeholder claims are common and require prioritization, which should be based on suitable categorization [27,40]. Relevant strategies are based either on the degree of involvement (internal and external) (met at a frequency of 20.05% in the relevant literature body studied herein) or on the degree of salience (power, legitimacy and urgency) (with a frequency of 21.35%), which lately varyingly modified to include the stakeholder influence methodology (at a frequency of 18.86% after 2015).

Community members loosely dependent on post-mine operations (such as entrepreneurs or farmers) are usually classified as demanding stakeholders with urgent claims (e.g., on the land exploitation) but with no power or legitimate relationship; on the other hand, the directly dependent community members (such as the mine workers) possess urgent and legitimate claims but have no power [36,37]. These two groups introduce two major complications that have puzzled quite a few researchers. The first complication has to do with the degree of stakeholder engagement with respect to stakeholder legitimacy (e.g., see [18,38]). The mine-dependent primary interested parties become the post-mine definitive stakeholders (i.e., those who will use the rehabilitated area as compensation for losing their jobs) with a given priority to their claims but without clear objectives, as they struggle through job changes. Within a stressed and disintegrated community, however, the demanding stakeholders are more likely to act and express their interest in project decisions, since the new reality favors the expansion of their economic activities [20,22,39]. Nonetheless, the latest views suggest that, especially when the community engagement or impact is expected to be extensive in size and duration, a more ethical and effective approach is required: *stakeholder management* should be replaced by *managing for stakeholders*, in the sense that all stakeholders have the right and legitimacy to receive attention for their claims (e.g., see [38,45]).

The second complication, frequently seen in large projects, comes from the inability of stakeholder management frameworks to handle potential changes in the stakeholder network. Post-mining projects, however, are characterized by frequent changes among stakeholders and their salience throughout the rehabilitation phases [37,43,45]; for example, many workers retire, while some locals differentiate their line of work to better exploit the reclaimed lands. Inevitably, each phase requires different contributions, uses different

process prioritizations and yields different results. Thus, stakeholder identification, classification and management should take place in each project phase, although this might prove to be complex in very large projects. Alternatively, social network analysis could give real-time insights into the evolving network, and it has been strongly suggested as a useful complement to existing stakeholder analysis methods (e.g., see [28,36,46]).

A significant part of the relevant published papers clearly stressed the need for effective planning of communication channels in order to ensure broader consensuses and stakeholder involvement, as both are considered critical to project success (e.g., see [17,23,28,36,39]). Surprisingly, there exists a considerable attempt to link stakeholder management to sustainability through project risk management and project performance assessment [40,41,47]; both functions necessitate some balancing between the economic, social and environmental project dimensions, thereby implementing sustainability principles. Communication is a key concept within this approach, facilitating the expansion of the stakeholders' networks and community-oriented prioritization in order to enhance the social performance of any project [36,39,47].

Another interesting finding in the present study refers to interorganizational collaborations that give rise to both temporary and permanent project networks. Rehabilitation projects require a certain degree of collaboration among public and private sectors (e.g., see [36,38,48]), resulting in the production of technological solutions for the benefit of society.

### 3.2.3. Land-Use Change

The land-use change node (Figure 2) showed one important association under the social perspective, i.e., the land-use pattern. Its most important aspect (centrality of 0.21 within the cluster) was the monitoring networks owing to the high contribution of reliable field data to mine reclamation success: most monitoring activities cease at mine closure leading to loss of monitoring data; yet, field data are required for the elucidation of the site-specific key closure hazards and risk drivers [25,49]. The monitoring need was also stressed with respect to the rehabilitation duration: closure planning involves long temporal scales, adequacy to accommodate changes in the regulatory framework, the social needs, the budget and funding planning, etc.; thus, the reliable real-time assessment of environmental quality is imperative when reengineering is sought for.

Policy drivers (centrality of 0.14 within the cluster) proved to be important to land-use patterns, especially in discussions about reclamation objectives and outcomes. Social drivers scored quite low (0.07), whereas technology drivers (centrality of 0.17 within the grid) arose mainly due to the restoration activities (Figure 4), aiming at pollution management in general and soil erosion in particular.

A significant association of restoration technology (not shown in Figure 2) was post-closure planning (centrality of 0.31 within the cluster); this topic structured a very comprehensive sub-cluster of related terms at lower levels of causality as per the factors contributing to failure in planning, judged by their frequency of occurrence in the relevant literature (Figure 5). Ecosystem services in post-mine areas cannot be expected to be similar to those existing before the exploitation of the mine since the degradation of the environment during mining—especially regarding surface and groundwater pollution, soil erosion, air quality and habitat loss or fermentation—is largely irreversible [50–52]. Thus, planning should be long-term so as to give time to the current ecosystem capacity to shift towards a more desirable state. Regional planning and land-use change tools are usually put in place to define the scope of rehabilitation and plan the new ecosystem services. Cost–benefit analysis is strongly proposed with a view to address critical environmental problems and to provide cost-efficient solutions (e.g., see [53–55]). Further, pre-mine and post-mine environment dependencies differ significantly in both the degree of dependence and the nature of dependence; for example, a pasture land serving a few community members before mining should be returned to the post-mine community as many viable and environmentally friendly alternatives for economic development.

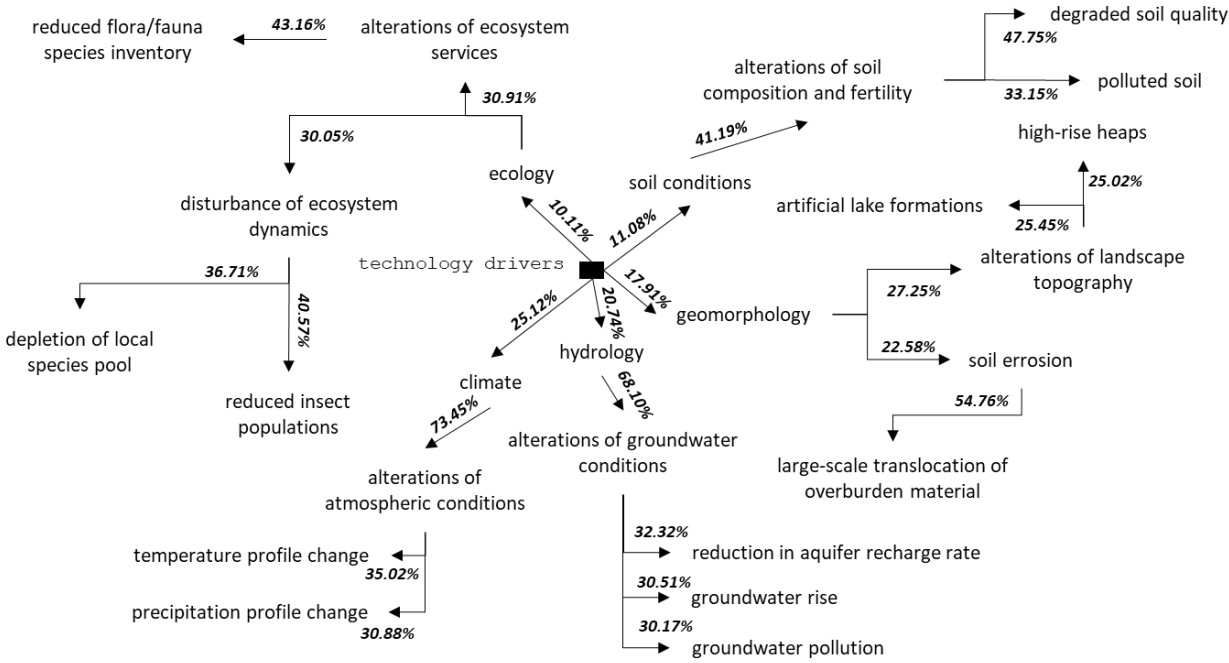

**Figure 4.** Top-down analysis of *technology drivers* in the *land-use change* node with respect to critical environmental challenges in mine rehabilitation as judged by their respective frequency of occurrence and their association with upper-level causal drivers (percentages in parentheses indicate the frequency of occurrence within the level).

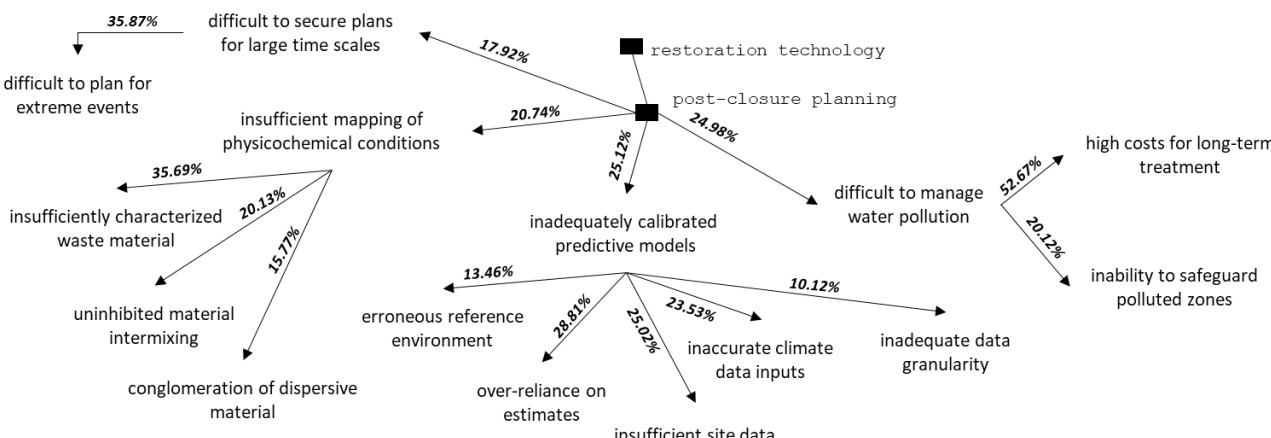

**Figure 5.** Top-down analysis of *post-closure planning*, linked to *restoration technology* in the *land-use change* node with respect to factors contributing to failure in planning as judged by their respective frequency of occurrence and their association with upper-level causal drivers (percentages in parentheses indicate the frequency of occurrence within the level).

Evidently, the scope of land repurposing is a serious challenge in its own right. In most cases (28.2%), forest reclamation is preferred, followed by agricultural reclamation (19.8%) and recreational reclamation (15.4%). Latest publications (after 2019) emphasize the need for synergies between recreational benefits, ecological functions, business activities and nature conservation (e.g., see [56–58]). In any case, the site-specific conditions should be carefully observed in planning the rehabilitation schemes [55,58], putting emphasis on the development of both productive and non-productive habitats.

### 3.2.4. Sustainability

Sustainability formed an extensive cluster accommodating a variety of inter-related concepts. A significant branch (centrality of 0.33 within the cluster) referred to ethics, mostly in the sense of resource saving, pollution abatement (environmental protection, impact mitigation and footprint reduction, in order of significance) and risk management. Environmental responsibility, as part of ethics, gave rise to an almost circular grid-linked environment through regulatory management and social responsibility; thus, these concepts are used inter-changeably by the scientific community.

Mine reclamation and just transition were linked to sustainability through three axes: mining activities, reclamation/rehabilitation activities and post-reclamation activities. Planning for the post-mine phase even before mining can help in both, tuning the mining operations to facilitate progressive land reclamation (e.g., increased operational efficiency, water management, energy saving, excavated material management, pit management, social responsibility, etc.) (e.g., see [59–61]) and increase sustainability indices through a life cycle view (e.g., see [62,63]. Further, the activities planned for rehabilitation should also follow sustainability principles [64–66], while the re-purposed area should be able to support a sustainable local economy [19,21,23,39,67].

Sustainability was also linked to sustainable reclamation (centrality of 0.17 within the cluster), mainly determined through regional planning, and sustainable integration (centrality of 0.22 within the cluster), mainly determined through circular economy. The encouragement of circular material use, through recycling and reuse, can save resources and minimize waste (e.g., see [68–71]). Still, some researchers feel that circular economy goals in mining are somewhat elusive and mostly deal only with waste management [72–74]. Notwithstanding, in planning a mega-project on the basis of a circular economy, all the conditions of sustainability should be met, even if it is necessary to accept a weaker conception of sustainability with the consent of all stakeholders.

### 3.2.5. Limitations of the Study

It is worthwhile to note that the relative importance of each node described herein is linked to the period covered in this review. That is, it may soon be that with new works on the subject, the importance of some themes will disappear, and others will emerge as important. The methodology described herein entails, in effect, modeling of the current knowledge output, a process that cannot be fully automated without severely compromising the reliability of the results. Thus, the unavoidable trade-offs between the duration of the study, depth of analysis and extent of the network should be carefully managed.

Additionally, the results presented herein are derived from comprehensive academic studies that implemented theoretical considerations over just transition issues with limited field feedback. These findings need to be further verified by non-academic reports and actual long-term field studies deriving from affected/impacted groups (e.g., NGOs, community representatives, etc.) and from the project contributor groups (e.g., local government groups, financial contributors, etc.). Given the short-term experience of site reclamation, it is quite possible that the decisive factors would be site- and community-specific, and they could entail other issues that have been underestimated herein, e.g., investments.

### 4. Conclusions

Mine reclamation is a long-term and complex technology-based process. The socioeconomic implications arising from such activities are equally long-term and complex. Using an AHP-based roadmapping methodological framework in order to screen the latest publications, this paper presented the critical areas that the researchers indicate as policymaking drivers. Although these findings need to be further verified by non-academic reports and actual long-term field studies, the issues highlighted herein would most certainly facilitate the just transition of policymaking to a more sustainable basis.

Notwithstanding, this study revealed that some major obstacles to the fair development transition of lignite areas derive from the social implications that mine closure brings

about. Over the many years of mine operation, a mine-dependent community develops a culture and an economy that are difficult to change. It seems that the relocation of the workers is not a preferred strategy, necessitating a reconfiguration of the whole community. Most data suggest that the preparation of the community for mine closure lacks in both the right timing and suitable strategy. Further, the rehabilitation is definitely going to be long-term, a fact that raises costs, wears down society and increases decommissioning timelines.

In effect, the social environment requires much attention. The early preparation of the community along with the strategic long-term development of alternative livelihood means, preferably through the rehabilitated areas, seem to be critical keys to a just transition. Appropriate training programs and targeted financial support to the mine workers might be proven adequate to lessen community disintegration. Further, holistic and realistic planning should be put forward with a view to gradually change the local prospects so as to spare high rehabilitation costs and unfinished projects.

Stakeholders' management in post-lignite projects poses significant challenges that actually go beyond divergent interests and rivalries. Using the stakeholder influence perspective, the appropriate handling of the strongly dependent and the loosely dependent parties proves difficult, especially at prioritizations. While the former struggle to adapt to their new life prospects and are scarcely prepared to plan ahead and get involved in post-mining activities, the latter are eager to seize new business opportunities. Further, stakeholder salience changes with time, thus necessitating different management processes at every phase of the procedure. Having a considerable duration, the rehabilitated area will be delivered to a different body of stakeholders, possibly with different scopes and intentions than the body of stakeholders at mine closure. Thus, broader consensuses and stronger links to regional planning seem to be indispensable.

As it was stressed out herein, the lack of long-term monitoring data presents a serious threat to mine reclamation success. The biggest problems reported from mine operations are pollution, waste management and soil erosion; their management post-mine is very demanding and requires real-time field data. In addition, land-use changes cannot be successful unless planned on and prioritized over an ecosystem services basis, which also requires extensive monitoring. The incorporation of ecology principles in restoration and the fostering of natural habitats in post-mine areas is strongly suggested for strengthening site-specific environmental quality; yet, the coupling of such planning with the development of an alternative economy might not be feasible.

Sustainability has a strong context to energy transition which is, however, very difficult to grasp in all its dimensions. Environmental responsibility and risk management are commonly employed as acceptable strategies. Still, they are not adequate to support life cycle thinking policies. The concepts entailed in sustainability are indeed in agreement with fair development, but they should be better clarified, especially with respect to circular economy targets.

**Funding:** This research received no external funding.

**Institutional Review Board Statement:** Not applicable.

**Informed Consent Statement:** Not applicable.

**Data Availability Statement:** The data presented in this study are available on request from the corresponding author. The data are not publicly available due to privacy restrictions.

**Acknowledgments:** The author would like to thank the Special Issue Editors.

**Conflicts of Interest:** The author declares no conflict of interest.

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
