# Peer review of "Fair Development Transition of Lignite Areas: Key Challenges and Sustainability Prospects"

_sustainability, doi:10.3390/su151612323_

Round 1
Reviewer 1 Report
It is recommended to modify Figure 2 and Figure 3 to highlight the sense of hierarchy by changing the color and shape of key nodes.
Author Response
Reviewer #1
|
Reviewer’s Comments |
Author’s response |
Changes in text |
|
|
The author would like to thank the reviewer for his/her positive opinion. In this work, the author used a structured approach to exploring the justified opinion of scientists and experts on the barriers to the just transition, as these emerge from the relevant studies on the dynamic development of the research field. |
|
|
1. It is recommended to modify Figure 2 and Figure 3 to highlight the sense of hierarchy by changing the color and shape of key nodes. |
Suggestion accepted. Both Figures have been color coded in the revised manuscript in order to better show the hierarchical relations. |
See Figures 2 and 3 and the green highlighted part in the respective legends. |
Reviewer 2 Report
In the conclusion, another critical point regarding the methodology used was missing: The relative importance of each node is linked to the period covered by the review. That is, it may be that soon, with new works on the subject, the importance of some themes will disappear and others will emerge as important.
Author Response
Reviewer #2
|
Reviewer’s Comments |
Author’s response |
Changes in text |
|
|
The author would like to thank the reviewer for his/her positive opinion. In this work, the author used a structured approach to exploring the justified opinion of scientists and experts on the barriers to the just transition, as these emerge from the relevant studies on the dynamic development of the research field. |
|
|
1. In the conclusion, another critical point regarding the methodology used was missing: The relative importance of each node is linked to the period covered by the review. That is, it may be that soon, with new works on the subject, the importance of some themes will disappear and others will emerge as important. |
The author agrees with the reviewer. A brief comment has been added in the last section of Results and Discussion. |
See the cyan highlighted part at the Section 3.2.5 |
Reviewer 3 Report
The study explores the challenges associated with the transition of lignite areas towards fair development practices and the potential sustainability prospects. Overall, it addresses an important topic, provides valuable insights, and adds to existing knowledge. However, there are specific points that require careful attention and improvement before the manuscript can be considered for publication.
A network was employed to conduct the analysis. However, the underlying concept of this method and its potential to yield reliable results within the context of the modified analytical hierarchy process-based roadmapping methodological framework were not presented in either the introduction or the materials and methods.
The statement “Although the results of this study need to be further verified by non-academic field players, they could facilitate the just transition to a more sustainable post-mine future” is indeed unclear, particularly in defining “non-academic field players.” To improve clarity, the authors should provide a more explicit explanation of who these “non-academic field players” are and their role in the verification process. Additionally, the statement is inadequately placed. It would be more appropriate to include this information in the discussion as a limitation of the study.
L88 How was the network constructed for this study, and what software, along with its version, was utilized in the network building process?
L137 In Figure 2, certain lines are continuous, while others are dashed. The caption should clearly describe the manner in which the results are presented.
L354 Incorporating the implications of the results, the limitations of the study, and the reliability of the methods into the discussion is indispensable for creating a thorough and transparent research paper.
Minor editing of English language required
Author Response
Reviewer #3
|
Reviewer’s Comments |
Author’s response |
Changes in text |
|
1. The study explores the challenges associated with the transition of lignite areas towards fair development practices and the potential sustainability prospects. Overall, it addresses an important topic, provides valuable insights, and adds to existing knowledge. |
The author would like to thank the reviewer for his/her positive opinion. Indeed, energy transition presents some critical uncertainties, especially as regards public acceptance and regional economy shifting. In this work, the author used a structured approach to exploring the justified opinion of scientists and experts on the barriers to the just transition, as these emerge from the relevant studies on the dynamic development of the research field. |
Not applicable |
|
2. However, there are specific points that require careful attention and improvement before the manuscript can be considered for publication. |
The author did her best to accommodate the suggestions of all reviewers in the revised manuscript and would like to thank the reviewers that helped improve the quality of the manuscript. |
Nor applicable |
|
3. A network was employed to conduct the analysis. However, the underlying concept of this method and its potential to yield reliable results within the context of the modified analytical hierarchy process-based roadmapping methodological framework were not presented in either the introduction or the materials and methods.
|
The author agrees with the reviewer that the methodology should be better explained. A brief presentation is now included in the beginning of the Methodology Section. |
See gray highlighted text at the first two paragraphs of methodology. |
|
4. The statement “Although the results of this study need to be further verified by non-academic field players, they could facilitate the just transition to a more sustainable post-mine future” is indeed unclear, particularly in defining “non-academic field players.” To improve clarity, the authors should provide a more explicit explanation of who these “non-academic field players” are and their role in the verification process. Additionally, the statement is inadequately placed. It would be more appropriate to include this information in the discussion as a limitation of the study. |
Suggestion accepted. A new Section has been added in Results and Discussion: 3.2.5 Limitations of the Study, where the importance of long-term feedback from the interested parties is discussed. |
See gray highlighted text at Section 3.2.5 |
|
5. L88 How was the network constructed for this study, and what software, along with its version, was utilized in the network building process? |
All software used herein was homemade. This is clearly indicated at the second paragraph of methodology. The network building process is briefly described at Section 2.2. |
See the last sentence at the second paragraph of Methodology and the gray highlighted text at the first paragraph of 2.2 |
|
6. L137 In Figure 2, certain lines are continuous, while others are dashed. The caption should clearly describe the manner in which the results are presented. |
Suggestion accepted. The legend of Figure 2 is more detailed in the revised manuscript. |
See gray highlighted text in the legend of Figure 2. |
|
7. L354 Incorporating the implications of the results, the limitations of the study, and the reliability of the methods into the discussion is indispensable for creating a thorough and transparent research paper. |
Suggestion accepted. The limitations of the study and reliability issues are discussed in section 3.2.5 of the revised manuscript. The implications of the study are discussed in the expanded Conclusions of the revised manuscript. |
See Section 3.2.5, the gray highlighted text at the first paragraph of Section 3.2.5 and the yellow highlighted text in Conclusions |
|
8. Minor editing of English language required |
Suggestion accepted. The revised manuscript has been thoroughly checked for language. |
See revised text. |
Reviewer 4 Report
The article has an interesting and valuable topic. There are many lignite mines in Europe that will be liquidated. This is a major environmental and social challenge. In this context, the article also corresponds to the subject of the journal. The authors also carry out a thorough review of the literature. They use the right methodology for this.
The text is clear, but it misses some important elements. First of all, in the end there should be a list of challenges, because this is the purpose of the article. Meanwhile, the conclusions are just a few lines.
As for the literature, it is up-to-date, but the list is very sparse for a review article. It must be completed. The results can be duplicated, the methodology is clear and well described.
The illustrations are precise and well developed.
Author Response
Reviewer #4
|
Reviewer’s Comments |
Author’s response |
Changes in text |
|
1. The article has an interesting and valuable topic. There are many lignite mines in Europe that will be liquidated. This is a major environmental and social challenge. In this context, the article also corresponds to the subject of the journal. The authors also carry out a thorough review of the literature. They use the right methodology for this. |
The author would like to thank the reviewer for his/her positive opinion. Indeed, energy transition presents some critical uncertainties, especially as regards public acceptance and regional economy shifting. In this work, the author used a structured approach to exploring the justified opinion of scientists and experts on the barriers to the just transition, as these emerge from the relevant studies on the dynamic development of the research field. |
Not applicable |
|
2. The text is clear, but it misses some important elements. First of all, in the end there should be a list of challenges, because this is the purpose of the article. |
Suggestion accepted. Concluding Remarks has been substituted with Conclusions in the revised manuscript, hosting a comprehensive and somewhat extensive discussion on the challenges to fair transition. |
See the yellow highlighted text in Conclusions |
|
3. Meanwhile, the conclusions are just a few lines.
|
Suggestion accepted. Concluding Remarks has been substituted with Conclusions in the revised manuscript, hosting a comprehensive and somewhat extensive discussion on the challenges to fair transition. |
See the yellow highlighted text in Conclusions |
|
4. As for the literature, it is up-to-date, but the list is very sparse for a review article. It must be completed. |
Suggestion accepted. The reference list has been expanded by 21.62% with the inclusion of 16 new references scattered within the chapters of the revised manuscript. |
The new additions are highlighted in yellow in the References section |
|
5. The results can be duplicated, the methodology is clear and well described. |
|
Not applicable |
|
6. The illustrations are precise and well developed.
|
|
Not applicable |
Round 2
Reviewer 4 Report
The authors corrected the article. I have no further suggestions. The paper could be accepted.